# ATRA influences the differentiation and fusion of myoblasts by regulating Rarα/Pitx2, leading to abnormal development of the pelvic floor muscles (PFMs) in fetal rats

Hanbin Zhao[2], Jian Cao[2], Huaqi Mu[2], Yang Bi[2], Zhenhua Guo[2], Yuan Shi[1]*, Yi Wang[2]*

**1** Department of Neonatology, National Clinical Research Center for Child Health and Disorders, Ministry of Education Key Laboratory of Child Development and Disorders, Chongqing Key Laboratory of Child Neurodevelopment and Cognitive Disorders, Children's Hospital of Chongqing Medical University, Chongqing, P. R. China, **2** Department of General Surgery &Neonatal Surgery, National Clinical Research Center for Child Health and Disorders, Ministry of Education Key Laboratory of Child Development and Disorders, Chongqing Engineering Research Center of Stem Cell Therapy, Children's Hospital of Chongqing Medical University, Chongqing, P. R. China

* shiyuan@hospital.cqmu.edu.cn (YS); ddxek1994@outlook.com (YW)

## Abstract

Anorectal malformation (ARM) is the most common congenital digestive tract anomaly in newborns, and children with ARM often have varying degrees of underdevelopment of the pelvic floor muscles (PFMs). To explore the effects of Rarα (NR1B1) and Pitx2 on the development of rat PFMs, we constructed a rat ARM animal model using all-trans retinoic acid (ATRA), and verified the expression of Rarα and Pitx2 in the PFMs of fetal rats. Additionally, we used rat myoblasts (L6 cells) to investigate the regulatory roles of Rarα and Pitx2 in skeletal muscle myoblast differentiation and their interactions. The results indicated a significant decrease in the expression of Rarα and Pitx2 in the PFMs of fetal rats with ARM. ATRA can also decrease the expression of Rarα and Pitx2 in the L6 cells, while affecting the differentiation and fusion of L6 cells. Knocking down Rarα in L6 cells reduced the expression of Pitx2, Myod1, Mymk, and decreased myogenic activity in L6 cells. When Rarα is activated, the decreased expression of Pitx2, Myod1, and Mymk and myogenic differentiation can be restored to different extents. At the same time, increasing or inhibiting the expression of Pitx2 can counteract the effects of knocking down Rarα and activating Rarα respectively. These results indicate that Pitx2 may be downstream of the transcription factor Rarα, mediating the effects of ATRA on the development of fetal rat PFMs.

## Introduction

Anorectal malformation (ARM) is the most common congenital digestive malformation, with an incidence rate of approximately 1/5000 [1]. With the improvement of surgical techniques and postoperative management, most of patients can achieve

---

---

**Data availability statement:** All relevant data are within the paper and its Supporting information files.

**Funding:** This study was supported by the Program for Youth Innovation in Future Medicine, Chongqing Medical University (grant No. W0125).

**Competing interests:** The authors have declared that no competing interests exist.

a positive functional prognosis. However, some children may still experience fecal incontinence or constipation [2,3]. Abnormal development of pelvic floor muscles (PFMs) is an important influencing factor for postoperative defecation dysfunction in ARM patients. Currently, research on the abnormal development of PFMs in children with ARM is limited and primarily focuses on observing related phenotypic differences, without further exploration of the underlying mechanisms [4–6].

The formation of skeletal muscles requires mononuclear myoblasts to exit the cell cycle and fuse with each other to form new multinucleated myotubes. Then, the newly formed myotubes undergo further cell fusion to develop into mature myotubes. Transmembrane protein 8c (Tmem8c), also known as Myomaker (MYMK), is essential for myoblast fusion. It is transiently expressed and effectively promotes myoblast fusion during myogenesis and muscle regeneration processes [7]. MyoD1 is a crucial transcription factor in myogenesis, playing a significant role in determining myoblast fate and regulating the transcription of the majority of muscle-specific genes [8]. Myod1 can directly bind to the promoter of Mymk and induce its transcription during myoblast fusion [9,10]. At the same time, the initial activation of Myod1 requires Pitx2 to directly bind to the core enhancer of MyoD1 [11]. Pitx2 is a paired-related homeobox gene involved in the molecular processes that regulate embryonic and fetal muscle development [12]. It is specifically expressed in all MyoD+, Myf5+, and myogenin+ cells of embryonic muscle primordia and regulates the proliferation and differentiation of myogenic progenitor cells during myogenesis [13,14]. There is no relevant report on whether Pitx2 is involved in causing abnormal PFMs development in ARM fetal rats.

All-trans retinoic acid (ATRA) is the active form of Vitamin A in the body, and it plays an important role by binding to the retinoic acid nuclear receptor family (Rarα, Rarβ, Rarγ), and physiological levels of ATRA are crucial for maintaining normal embryonic development [15]. Excessive ATRA can cause various malformations, including ARM, neural tube defects, and cleft palate [16]. In our previous study, we summarized a modeling method for ARM with a success rate as high as 97% by testing different concentration gradients and administration times [17].

Rarα can promote the maturation and differentiation of myogenic progenitor cells [18]. Rarα is reduced in the terminal rectum of patients with ARM, while the expression of Rarβ and Rarγ shows no significant abnormalities [19]. Previous studies have reported that there is no Rarα transcription factor in the distal colon region of fetal rats treated with excessive ATRA, and the impairment of the Rarα signal pathway is one of the reasons for the interference of ATRA in the development of the rat caudal region [20]. However, the regulatory role of Rarα in the myoblasts of the fetal rat pelvic floor has not been confirmed. The downstream molecular mechanism of Rarα's impact on the development of rat caudal region tissues has not been clarified yet.

This study aims to explore the impact of ATRA on fetal rat PFMs and investigate the expression of Rarα and Pitx2 in the PFM tissues of fetal rats with anorectal malformation. In addition, we used rat myoblasts L6 as a model to investigate the correlation between Rarα and Pitx2. This study aims to provide new insights into the molecular mechanisms underlying the abnormal development of PFMs in children with ARM.

## Materials and methods

### Animal and housing

The Sprague-Dawley (SD) rats (weighing 200–220g) were purchased from the Animal Experiment Center of Chongqing Medical University and were housed in the Experimental Animal Center of Chongqing Medical University Affiliated Children's Hospital. Male and female rats were housed together in a 1:1 ratio, and the day on which a vaginal plug was observed was designated as embryonic day 0.5 (E0.5). The experimental group used 3% isoflurane inhalation until the rats stopped moving, ATRA at a dose of 80 mg/kg was administered by gavage on E10.5, while the control group received the same amount of corn oil by gavage. Rats were euthanized by passive exposure to $CO_2$ for 15 min. Pregnant rats were euthanized and fetal rats were collected on E16.5 and E17.5 (supplement Table 1). The animal use protocol has been reviewed and approved by the Ethics Committee of Children's Hospital of Chongqing Medical University (CHCMU-IACUC20210114038).

### Cell culture and differentiation

L6 cells were cultured in Dulbecco's Modified Eagle's Medium (DMEM; Gibco, Grand Island, NY) containing 10% fetal bovine serum (Gibco, USA), 100 IU/ml penicillin, and 100 IU/ml streptomycin. The cells were plated at approximately 30% fusion degree, allowed them to attach in complete culture medium, and cultured in differentiation medium (DM) containing 2% horse serum (Solarbio, China) for 24 hours before adding stimuli [21]. For immunofluorescence experiments, the cells were plated at 20% confluence and allowed to grow overnight before switching to DM medium for further processing.

### Inhibition and activation of Rarα and Pitx2 expression

The Rarα-specific agonist AM580 (Abmole, USA) was used to activate Rarα, and the adenovirus used to inhibit Rarα expression was provided by Associate Professor Bi Yang from the Stem Cell Laboratory, Chongqing Medical University Affiliated Children's Hospital [22] (supplement Fig 1). The primer sequences can be found in Table 1.

Plasmids were utilized to suppress the expression of Pitx2, and transfection was assisted by the cationic transfection reagent Lipofectamine 2000 (Invitrogen, USA). L6 cells were transfected with the plasmid using Lipo 2000 and incubated for 48 hours. Three pairs of shRNA genes were designed to target three gene loci, and the specific oligonucleotide sequences can be found in Table 2. The results indicated that rPitx2-shRNA2 was the most effective in reducing Pitx2 expression in L6 cells, so it was selected for subsequent experiments (supplement Fig 2). Ad-Pitx2 adenovirus was purchased from Tsingke Biotech. After transfection with adenovirus, the cells were switched to DM medium containing ATRA (10 µM) or AM580 (20 nM). After treating with ATRA or AM580 for 48 hours, proteins were extracted for the experiment, and each group was repeated at least 4 times. Cells were cultured in DM medium as the control group.

### qRT-PCR

Total RNA was extracted from rat pelvic floor tissue and L6 cells using TRIzol (Invitrogen, USA). The RNA was then reverse transcribed to cDNA using the PrimeScript™ RT Reagent Kit (Takara, Japan) according to the manufacturer's instructions. The primers used in the study are listed in Table 3, and RT-qPCR was performed using the CXF96 system (Bio-Rad, USA).

**Table 1. sequence of primers.**

| Name of primers | Sequence of primers |
| --- | --- |
| Rat Rarα siRNA site | 5'-aGCAAGTACACTACGAACAAtttt-3' |
| | 5'-aTTGTTCGTAGTGTACTTGCtttt-3' |

**Table 2. Nucleotide sequence of the small hairpin RNAs targeting Pitx2 mRNA.**

| Plasmid | Forward primer | Reverse primer |
|---|---|---|
| rPitx2-shRNA1 | GATCGCCTGAATAACTTGAACAACCCTC-GAGGGTTGTTCAAGTTATTCAGGC TTTTTT | AATTAAAAAAGCCTGAATAACTTGAACAAC-CCTCGAGGGTTGTTCAAGTTATTCAGGC |
| rPitx2-shRNA2 | GATCGCCTTTGATTTCAAAGGAATGCTC-GAGCATTCCTTTGAAATCAAAGGC TTTTTT | AATTAAAAAAGCCTTTGATTTCAAAGGAAT-GCTCGAGCATTCCTTTGAAATCAAAGGC |
| rPitx2-shRNA3 | GATCGCGAGCAAAGGAATGTATATACTC-GAGTATATACATTCCTTTGCTCGCTTTTTT | AATTAAAAAAGCGAGCAAAGGAAT-GTATATACTCGAGTATATACAT-TCCTTTGCTCGC |

**Table 3. Sequences of primes used for PCR.**

| Gene | Forward primer | Reverse primer |
|---|---|---|
| Rarα (rat) | CTTCCTCCAACCTGCACAACT | ATCCGTGCAAGTCGGCTAGA |
| Pitx2 (rat) | GGATCGTGGAAGTCGGCTC | GCAGACTCCAGCCAAACAGA |
| Gapdh (rat) | TAAAGGGCATCCTGGGCTACACT | TTACTCCTTGGAGGCCATGTAGG |

## Western blot analysis

Total proteins were extracted from rat pelvic floor tissue and cell samples using a total protein extraction kit (Beyotime, China), following the manufacturer's instructions. The proteins were separated by 12.5% SDS-PAGE and then transferred to a PVDF membranes (Millipore, USA). The membranes were then incubated with diluted primary antibodies against Rarα (1:800, ABclonal, China), Pitx2 (1:1000, Proteintech, China), Mymk (1:1000, ABclonal, China), Myod1 (1:1000, Proteintech, China), and GAPDH (1:1000, ZSGB-BIO, China) overnight at 4℃. Subsequently, the membranes were incubated with secondary anti-bodies (1:2000, Sizhengbo, China). The chemiluminescent signals were detected using a CEL detection kit (Bio-Rad, USA). Use the G: BOX chemical luminous imaging system (Synoptics, UK) for image collection and band density analysis.

## Immunofluorescence assay

For cell immunofluorescence staining, cells on cell-attached slides were fixed with 4% paraformaldehyde for 20 minutes and then treated with 0.3% Triton X-100 (Sigma, Ireland) for 20 minutes to increase cell permeability. Then, the cells were incubated with PBST containing 5% bovine serum albumin (BSA, Sigma-Aldrich, USA) for 25 minutes to block non-specific binding, followed by incubation with diluted primary antibodies against Mymk (1:200, ABclonal, China) and Myod1 (1:200, Invitrogen, USA) at 4℃ overnight. The corresponding secondary antibodies (1:200, chicken anti-mouse IgG Alexa Fluor 488 or chicken anti-rabbit Alexa Fluor 594, Invitrogen, USA) were subsequently incubated for 1 hour. The cells were then stained with DAPI for 40 minutes to visualize the cell nuclei. The slides were mounted with an anti-fluorescence quenching agent (Abcam, USA) and observed and photographed using a C2＋confocal microscope (Nikon, Japan). Differentiation index was measured as the percentage of the nuclei number in Myod1＋cells relative to the total nuclei number. Fusion index was measured as the nuclei number in Mymk+ (three or more nuclei) as a percentage of the total number of nuclei.

For histological analysis of fetal rat pelvic floor tissues, samples were embedded in OCT compound after immersion in 20% sucrose, and then frozen. The frozen tissues were cut into 10μm thick sections, which were then incubated in PBS containing 0.3% Triton X-100 (Sigma, Ireland) for 25 minutes. Subsequently, the sections were blocked with 5% BSA for 1 hour and incubated with mixed primary antibodies, including Pax7 (1:100, Santa Cruz, USA) + Pitx2 (1:200, Bioss, China), Rarα (1:200, Abcam, USA) + Pitx2, Pax7＋Mymk (1:200, ABclonal, China), and Pax7＋Myod1 (1:200, ABclonal, China) at 4℃ overnight. The sections were then washed and incubated with corresponding secondary antibodies (1:200, chicken

anti-mouse IgG Alexa Fluor 594 and chicken anti-rabbit Alexa Fluor 488, Invitrogen, USA) for 1 hour at room temperature, followed by DAPI staining for 15 minutes. After sealing with an anti-fluorescence quenching agent, the sections were observed and photographed using a fluorescence microscope (Nikon, Japan).

### Statistical analysis

The differentiation rate of L6 cells was determined by calculating the percentage of nuclei in differentiated myotubes (Myod1-positive cells) out of the total number of nuclei. Similarly, the fusion rate of myotubes was determined by calculating the percentage of nuclei in fused myotubes (Mymk-positive cells) out of the total number of nuclei. Image J software (National Institutes of Health, Bethesda, MD) was used to calculate the grayscale values of protein bands in Western blots, and GraphPad Prism 9.5.1 (GraphPad Software, La Jolla, CA) was used for data analysis. All data are presented as means ± standard deviations. The data with a normal distribution were analyzed using analysis of variance (ANOVA) with Tukey's multiple comparison test. P values less than 0.05 were considered statistically significant.

## Results

### Poor development of the PFMs in ARM fetal rats

In the normal group of fetal rats, the horizontal section stained with HE revealed a clearly visible rectum and urethra, as well as a symmetrical distribution of muscle fibers on both sides of the rectum and urethra. In the abnormal group (treated with ATRA), there was an absence of an anus, and the distribution of PFMs fibers was disorganized and sparse, accompanied by abnormalities of the sacrococcygeal neural tube (supplement Fig 3 c-h).

### Expression of Pitx2, Rarα, and muscle-generating related genes in the PFMs of fetal rat

mRNA and protein levels of Pitx2 and Rarα were investigated in the PFMs of fetal rats at E16.5 and E17.5 (Fig 1a, b). qRT-PCR results showed that the expression of Pitx2 in ARM fetal rat PFMs at E16.5 and E17.5 was significantly lower than that in the normal group（wt）($P < 0.01$ and $P < 0.001$, respectively), and Rarα expression was lower in the ARM group than in the normal group ($P < 0.01$ and $P < 0.001$, respectively). Western blot analysis indicated that Pitx2 and Rarα in the PFMs of the ARM group were also expressed at low levels (Fig 1c-e). Pax7 is a specific marker for myoblasts [23], and immunofluorescent staining was performed on horizontal sections of the pelvic floor. The fluorescence labeling of Pax7 was primarily concentrated in the muscle tissue area. Co-localization analysis revealed that the fluorescence intensity of Pitx2 and Rarα in the ARM group PFMs was significantly lower than that in the control group (Fig 1f), but the Pax7 signal was significantly stronger than in the control group (Fig 1g). The expression of the differentiation-related gene Myod1 and the fusion-related gene Mymk was also significantly decreased (Fig 1h, i).

### ATRA affects the differentiation and fusion of L6 and the expression of Rarα and Pitx2

Stimulation of L6 cells with 10μM ATRA in DM medium significantly decreased the expression of Rarα and Pitx2 ($P < 0.001$ and $P < 0.05$, respectively) (Fig 2a, b). Western blotting showed the same results as qRT-PCR (Fig 2c-e). AM580 (20nM), a specific agonist of Rarα, increased the expression of Pitx2 in L6 cells co-stimulated with ATRA compared to cells stimulated with ATRA alone ($P < 0.05$). It was also observed that ATRA significantly decreased the expression of Myod1 and Mymk in L6 cells ($P < 0.001$ and $P < 0.05$, respectively), while AM580 rescued the effect of ATRA on the expression of Myod1 and Mymk ($P < 0.0001$ and $P < 0.05$, respectively) (Fig 2f-h).

To investigate the impact of Pitx2 on the differentiation of L6 cells, we suppressed the expression of Pitx2 using shRNA (rPitx2-shRNA), and upregulated the expression of Pitx2 by transfecting L6 cells with Ad-Pitx2.

In DM medium containing ATRA, transfection of Ad-Pitx2 could only rescue the decrease in Myod1 ($P < 0.001$), but not the decrease in Mymk (Fig 2f-h). Cell immunofluorescence yielded the same results (Fig 3a). The fusion and

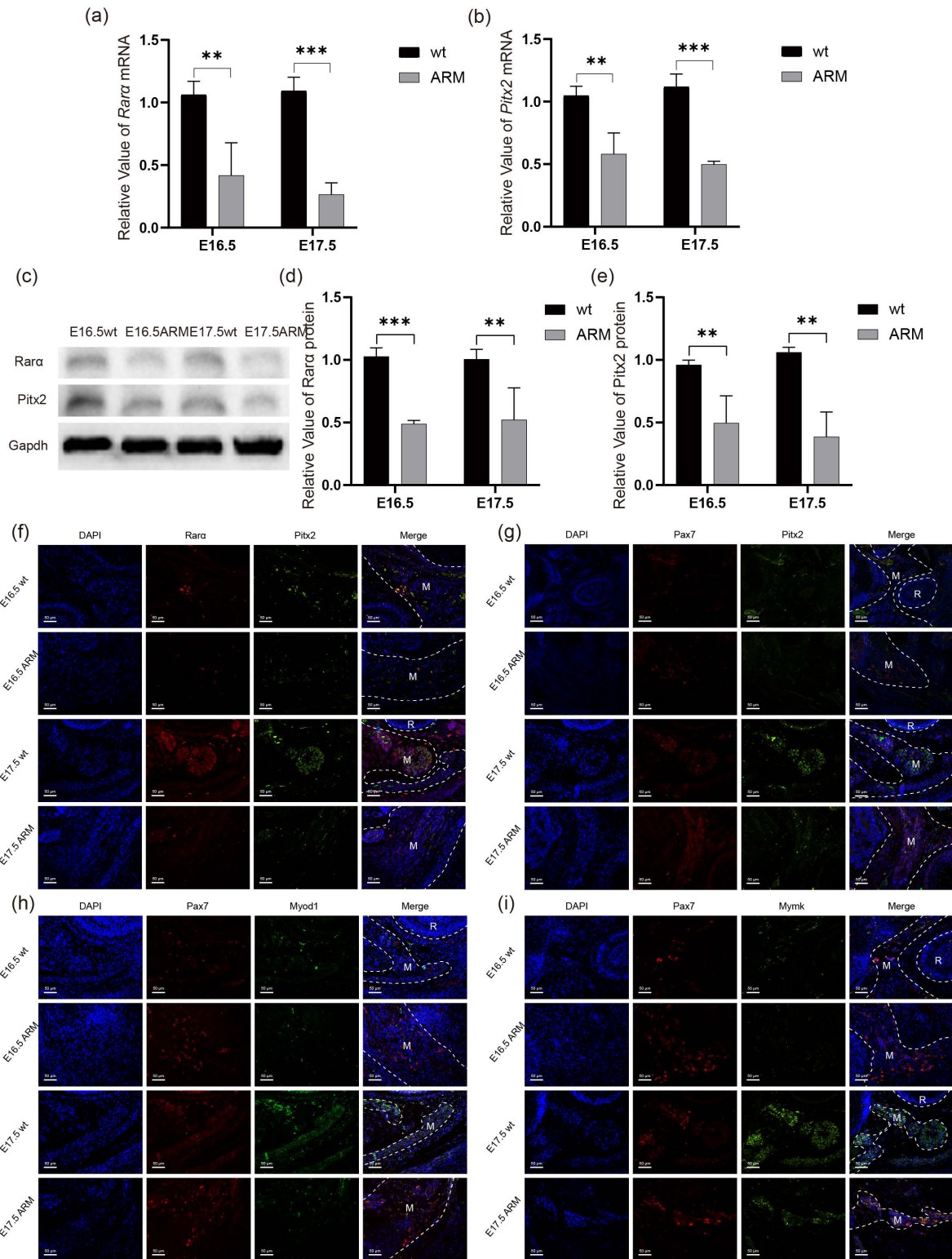

**Fig 1. Expression of Rarα and Pitx2 in fetal rat PFMs.** White dashed lines indicate muscle(M) or rectal(R) areas. **(a)** Rarα mRNA expression. **(b)** Pitx2 mRNA expression. **(c)** Western blot analysis shows Rarα and Pitx2 expression in PFMs (n = 4). **(d, e)** Quantification of Rarα (d) and Pitx2 (e) protein expression in **(c)**. Immunofluorescent staining for Rarα and Pitx2 **(f)**, Pax7 and Pitx2 (g), Pax7 and Myod1 (h), Pax7 and Mymk (i) in PFMs of fetal rats (magnification, 200×).

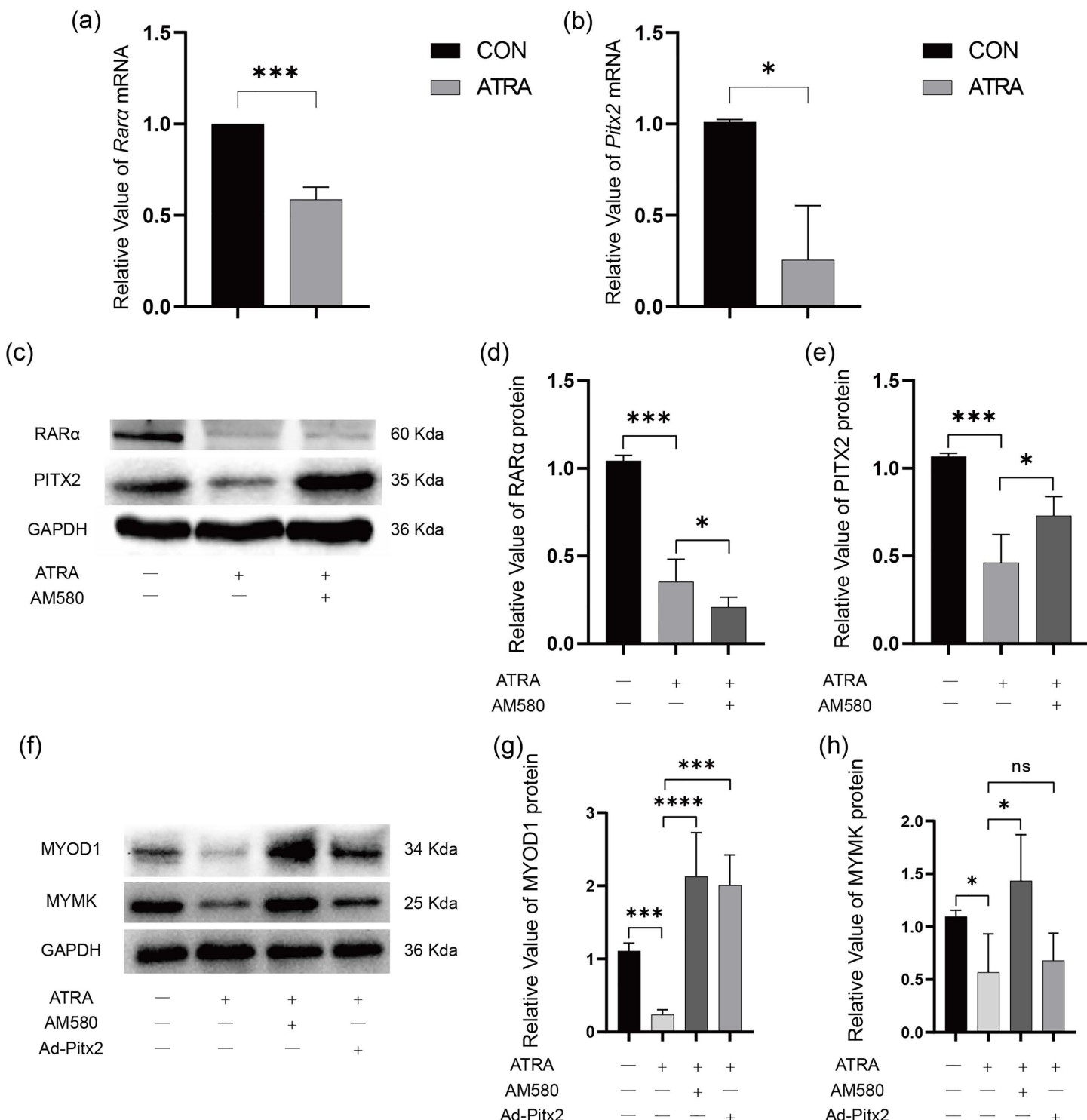

**Fig 2. ATRA affects the expression of Rarα and Pitx2 in L6 cell. (a, b)** Effects of ATRA on Rarα (a) and Pitx2 (b) mRNA expression of L6 cells. **(c)** Western blotting analysis of Rarα and Pitx2 protein expression after ATRA and AM580 stimulation. **(d, e)** Quantification of Rarα (d) and Pitx2 (e) in western blot shown in **(c)** (n = 4). **(f)** Western blotting analysis of Myod1 and Mymk protein expression after transfection with Ad-Pitx2 followed by treatment with ATRA or AM580. **(g, h)** Quantification of Myod1 (g) and Mymk (h) in western blot shown in **(f)** (n = 4).

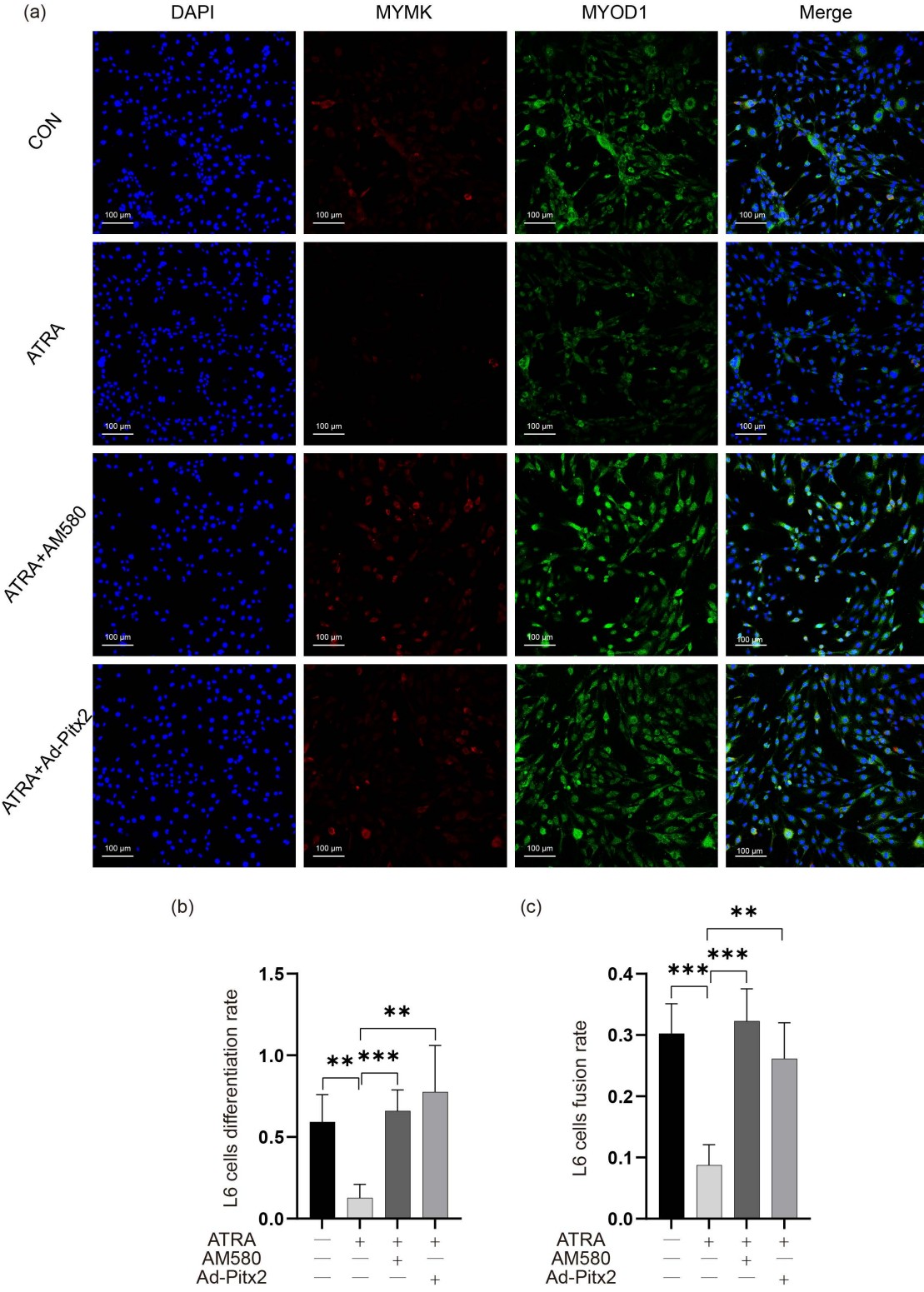

**Fig 3. The genes associated with differentiation and fusion in L6 cells were detected using immunofluorescence. (a)** Immunofluorescent staining for Mymk and Myod1 after ATRA, AM580 stimulation and Ad-Pitx2 infection (magnification, 200×). Quantification of the myotube differentiation rate **(b)** and fusion rate **(c)**.

differentiation rates of L6 cells were calculated based on co-expression analysis (Fig 3b, c). The differentiation rates of the control group, ATRA group, ATRA+AM580 group, and ATRA+ Ad-Pitx2 group were $0.592 \pm 0.167$, $0.127 \pm 0.082$, $0.659 \pm 0.129$, and $0.776 \pm 0.285$, respectively. ATRA significantly decreased the differentiation rate of L6 cells ($P < 0.01$), while AM580 and Ad-Pitx2 rescued the decreased differentiation rate caused by ATRA ($P < 0.001$ and $P < 0.01$, respectively). The fusion rates of the groups were $0.302 \pm 0.049$, $0.087 \pm 0.033$, $0.323 \pm 0.053$, and $0.261 \pm 0.059$, respectively. Similarly, ATRA significantly decreased the fusion rate of L6 cells ($P < 0.001$), and AM580 and Ad-Pitx2 rescued the decreased fusion rate caused by ATRA ($P < 0.001$ and $P < 0.01$, respectively). These results suggest that ATRA may inhibit the myogenic differentiation of L6 cells by downregulating Rarα and Pitx2.

### Rarα may affect the expression of Myod1 and Mymk through Pitx2 in L6 cells

The Ad-siRarα adenovirus was used to infect L6 cells to construct L6 cells with downregulated Rarα expression. After downregulating Rarα expression, it was observed that Pitx2, Myod1, and Mymk were all simultaneously decreased ($P < 0.0001$, $P < 0.001$, and $P < 0.0001$, respectively). The addition of AM580 after transient transfection with Ad-siRarα, or co-transfection of Ad-siRarα with Ad-Pitx2, can partially restore the decrease in Pitx2 ($P < 0.05$ and $P < 0.05$, respectively) and Myod1 ($P < 0.01$ and $P < 0.01$, respectively) resulting from Rarα downregulation (Fig 4a, 4c–4f). Stimulation of L6 cells with AM580 alone also increased the expression of Myod1 ($P < 0.01$) and Mymk. Simultaneously transfecting rPitx2-shRNA while stimulating L6 cells with AM580 can significantly attenuate the enhanced effect of AM580 on the expression of Myod1 and Mymk ($P < 0.001$ and $P < 0.05$, respectively) (Fig 4b, 4g, 4h).

## Discussion

In this study, we confirmed the abnormal development of PFMs in ARM rats, which is consistent with clinical observations. Furthermore, the expression of Rarα and Pitx2 in the PFMs of fetal rats with ARM was decreased. Based on these results, we conducted in vitro experiments and demonstrated that ATRA can affect the myogenic differentiation of rat L6 cells by decreasing the expression of Rarα and Pitx2.

In ARM patients, there are abnormalities in the development and distribution of PFMs, which worsen with the severity of ARM [5,24]. To investigate the mechanisms underlying these developmental abnormalities, we established an ARM model by administering ATRA to pregnant rats. No normal anal opening was observed in the model rats, and a rectourethral fistula was clearly observed in sagittal sections (supplement Fig 3 a, b), indicating the successful modeling of ARM. ATRA plays an important role in the development and functional maintenance of organs, and Rarα is one of its nuclear receptor family members and is present in many tissues, including muscles [25]. There is evidence that increased expression of Rarα can promote the differentiation and fusion of myoblasts [26]. The biological effects of ATRA are highly concentration-dependent. Excessive exposure to ATRA can not only downregulate RARα expression levels but also promote the ubiquitination and degradation of RARα protein by activating the proteasome pathway, thereby exerting negative feedback inhibition on the signaling pathway [18,27,28]. The results of this study revealed abnormal development of PFMs in ARM fetal rats. The expression of Rarα decreased, while immunofluorescence results indicated an increase in Pax7 expression, a marker of myoblasts [29]. Therefore, it is speculated that the effect may be attributed to the action of ATRA on the pelvic floor, leading to a reduction in Rarα in myoblasts, which hinders their ability to differentiate and fuse properly into mature muscle tissue. Knopp et al. demonstrated that Pitx2 promotes myogenic differentiation of myoblasts but reduces their proliferation [30]. Deletion of Rarα in periocular mesenchyme results in decreased expression of Pitx2. Additionally, specific inactivation of Pitx2 leads to abnormalities in periocular development and the maintenance of the optic nerve, similar to the eye defects observed in Rarα knockout rats [31,32]. During the development of PFM, it is unclear whether Rarα and Pitx2 exist in linear genetic pathways. Therefore, we measured the expression of Pitx2, and Mymk, a gene associated

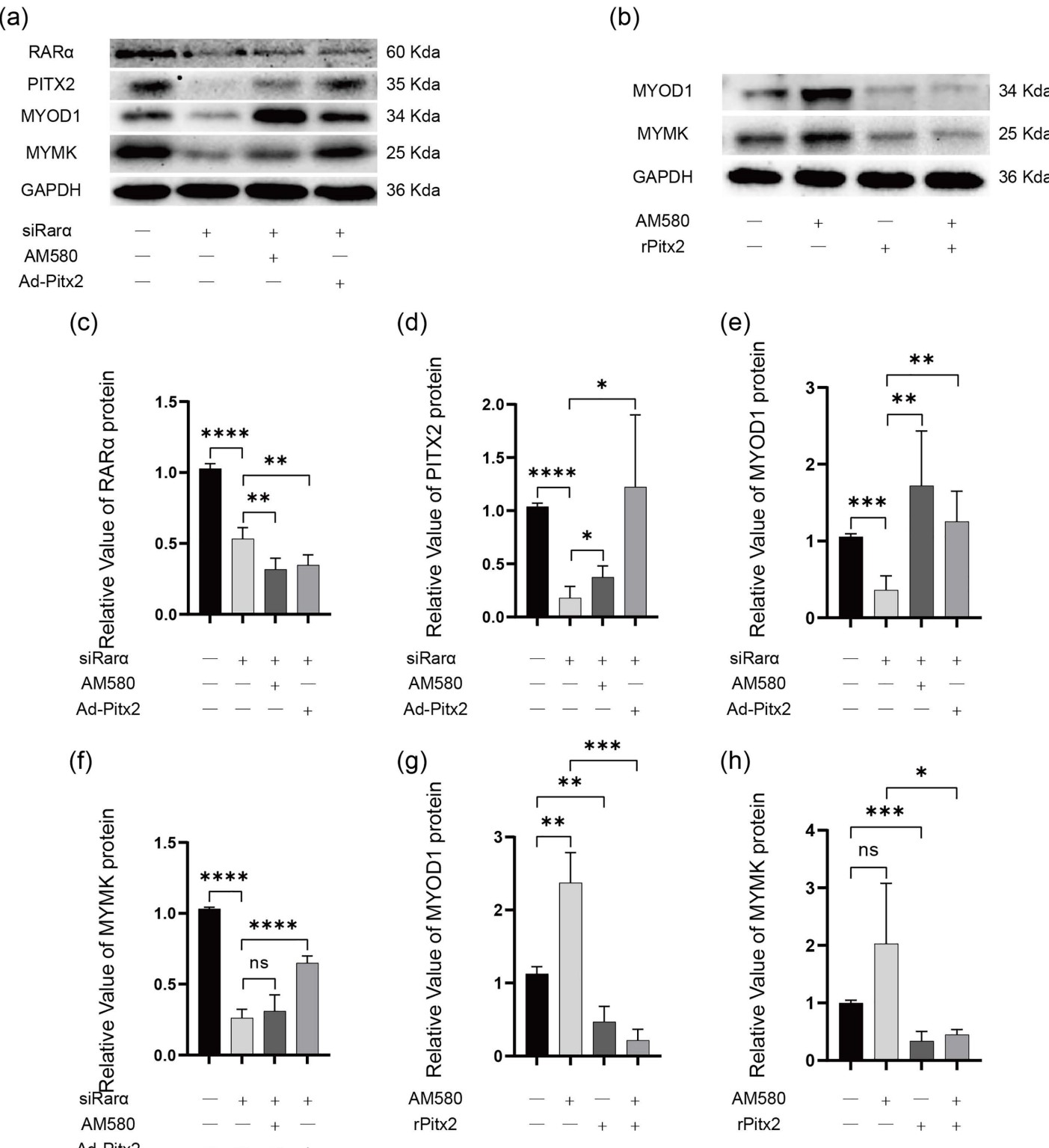

**Fig 4. Rarα affect the expression of Myod1 and Mymk through Pitx2. (a)** Western blotting analysis of Rarα, Pitx2, Myod1 and Mymk protein expression after transfection with Ad-siRarα and Ad-Pitx2 or followed by treatment with AM580. **(c-f)** Quantification of Rarα (c), Pitx2 (d), Myod1 (e) and Mymk (f) in western blot shown in (a) (n = 4). **(b)** Western blotting analysis of Myod1 and Mymk protein expression after AM580 treatment and rPitx2-shRNA plasmid transfection. **(g, h)** Quantification of Myod1 (g) and Mymk (h) in western blot shown in (n = 4) (b).

with myoblast fusion, and Myod1, a gene associated with differentiation [33,34]. The expression of these genes was significantly decreased in the model group.

To further clarify the effects of ATRA on myoblasts, we conducted in vitro experiments. Stimulation of L6 cells with ATRA in vitro significantly reduced the expression of Rarα and Pitx2, as well as the expression of Mymk and Myod1, which is consistent with the in vivo experiment. Our Western blot and immunofluorescence results indicate that the Rarα agonist AM580 or overexpression of Pitx2 can counteract the effects of ATRA on the differentiation and fusion of L6 cells. These results suggest that ATRA may influence the myogenic differentiation of L6 cells via Rarα or Pitx2.

In order to further investigate the impact of Rarα on L6 differentiation through Pitx2, we successfully achieved down-regulation and overexpression of target genes in L6 cells through RNA interference and exogenous plasmid transfection. By reducing the expression of Rarα, the expression of Pitx2, Mymk, and Myod1 decreased. While reducing Rarα, activating the remaining Rarα with AM580 can partially rescue the expression of Pitx2 and Myod1. The increase in Myod1 is greater than that of Pitx2. It is speculated that the reason for this may be that Pitx2 not only directly promotes Myod1 but also regulates the expression of myoblast prospecification markers such as Tbx1, Tcf21, and Msc. These transcription factors are also required for the activation of Myf5 and Myod1 [35]. Increased expression of Pitx2 can significantly enhance the expression of Mymk and Myod1, while decreasing the expression of Pitx2 can significantly impact the enhancing effect of AM580 on Mymk and Myod1. These results indicate that Pitx2 mediates the action of Rarα in regulating the myogenic differentiation of L6 cells. However, the mechanism of interaction between Rarα and Pitx2 is not clear. Some authors have performed computer analysis of the Pitx2 gene and did not find RarE(retinoic acid response elements) sequences in its promoter. Furthermore, microarray analysis did not reveal that Pitx2 is a target of Rarα [32,36]. Therefore, RA may indirectly regulate Pitx2 through transcription intermediaries. Activation of Rarα can lead to the initiation of intracellular signals triggered by TGF-β in cells [37]. After TGF-β binds to its receptor, the Smad protein bound to the intracellular domain of the receptor is phosphorylated and released. Subsequently, it acts as a transcriptional signal on target genes in the nucleus, leading to cell differentiation and growth inhibition [38]. Studies have shown that the Smad4 protein can bind to the Pitx2 promoter and stimulate the expression of Pitx2 [39]. Therefore, we speculate that the inhibition of L6 cell differentiation by reducing Rarα may be achieved through the TGF-β-Smad4 axis, resulting in reduced Pitx2 expression. In addition, this study also observed that even if Rarα is inhibited, AM580 can still increase the expression of Pitx2 and Myod1. There may be other nuclear transcription factors involved in regulating Pitx2 gene expression in L6 cells, which necessitates further experiments for verification.

In summary, our study results suggest that Rarα is essential for the differentiation of L6 cells, and its role in myogenesis may be mediated by Pitx2. ATRA may affect the development of PFMs in fetal rats through the Rarα/Pitx2 pathway. To further elucidate the mechanisms by which Rarα regulates myogenic differentiation, it would be useful to identify the mediators of Rarα and Pitx2 interactions. Further research is needed.

## Supporting information

**S1 Fig. Effects of various treatments on Rarα in L6 cells.** (a) L6 cells were induced to differentiate for various lengths of time. Western blotting analysis showing Rarαand Pitx2 expression at various stages. D0 indicates undifferentiated L6 cells, and D1 to D4 indicate L6 cells on Days 1–4, respectively, after initial exposure to differentiation medium. (b, c) Quantification of Rarαand Pitx2 protein expression in (a); (d, e) Differentiated medium (CON), vacuum virus, Rarα inhibitory adenovirus, and AM580 effect on Rarα; *p < 0.05, ***p < 0.001, ****p < 0.0001.
(DOCX)

**S2 Fig. The effect of different plasmids on Pitx2 expression in L6 cells.** (a, b) Among the three plasmids, sh2 exhibited the most significant inhibitory effect on Pitx2. (c, d) Differentiated medium (CON), vacuum virus, Pitx2 overexpression adenovirus, and blank plasmid, Pitx2 inhibitory plasmid effect on Pitx2. *p < 0.05, ***p < 0.001, ****p < 0.0001.
(DOCX)

**S1 Table. Comparison of the number of malformed fetal rats between the control group and the model group.**
(DOCX)

**S3 Fig. HE staining of fetal rat sections (magnification, 40×), R: rectum, A: anus, F: fistula, B: bladder, U: urethra, C: cloaca, Red dashed lines indicate muscle areas.** (a) Sagittal section of E17.5 fetal rats in the normal group. (b) Sagittal section of ARM group. (c, d, e) Horizontal sections of fetal rats at E16.5, E17.5, and E18.5 in the normal group. (f, g, h) Horizontal sections of fetal rats at E16.5, E17.5, and E18.5 in the ARM group.
(DOCX)

**S1 File. Original WB.**
(TIF)

## Author contributions

**Conceptualization:** Hanbin Zhao, Yuan Shi, Yi Wang.

**Data curation:** Hanbin Zhao, Jian Cao.

**Funding acquisition:** Yi Wang.

**Investigation:** Hanbin Zhao, Jian Cao, Huaqi Mu.

**Methodology:** Yang Bi, Yuan Shi.

**Project administration:** Zhenhua Guo, Yuan Shi, Yi Wang.

**Resources:** Yang Bi, Yi Wang.

**Supervision:** Yuan Shi, Yi Wang.

**Writing – original draft:** Hanbin Zhao.

**Writing – review & editing:** Yuan Shi, Yi Wang.

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
