## [Decision Letter · Decision Letter 0]

31 Jul 2025

Dear Dr. Wang,

We look forward to receiving your revised manuscript.

Kind regards,

Michael Schubert

Academic Editor

PLOS ONE

“This study was supported by the Program for Youth Innovation in Future Medicine, Chongqing Medical University (grant No. W0125).”

“This study was supported by the Program for Youth Innovation in Future Medicine, Chongqing Medical University (grant No. W0125).”

“This study was supported by the Program for Youth Innovation in Future Medicine, Chongqing Medical University (grant No. W0125).”

5. We note that you have indicated that there are restrictions to data sharing for this study. For studies involving human research participant data or other sensitive data, we encourage authors to share de-identified or anonymized data. However, when data cannot be publicly shared for ethical reasons, we allow authors to make their data sets available upon request. For information on unacceptable data access restrictions, please see http://journals.plos.org/plosone/s/data-availability#loc-unacceptable-data-access-restrictions.

7. Please include a separate caption for each figure in your manuscript.

8. We are unable to open your Supporting Information file [supplement.zip]. Please kindly revise as necessary and re-upload.

Reviewers' comments:

Reviewer's Responses to Questions

**Comments to the Author**

1. Is the manuscript technically sound, and do the data support the conclusions?

Reviewer #1: Partly

Reviewer #2: Yes

2. Has the statistical analysis been performed appropriately and rigorously?

Reviewer #1: Yes

Reviewer #2: Yes

3. Have the authors made all data underlying the findings in their manuscript fully available?

Reviewer #1: Yes

Reviewer #2: Yes

4. Is the manuscript presented in an intelligible fashion and written in standard English?

Reviewer #1: Yes

Reviewer #2: Yes

Reviewer #1: Tracking #: PONE-D-25-14533 – Hanbin Zhao et al.

The article entitled “ATRA influences the differentiation and fusion of myoblasts by regulating Rarα/Pitx2, leading to abnormal development of the pelvic floor muscles (PFMs) in fetal rats” presents some interesting new data concerning the mode of action of all-trans retinoic acid (ATRA) in anorectal malformation and in the development of the pelvic floor muscles.

Overall, this is a well-constructed manuscript. The introduction gives a good understanding of the subject and its issues, and provides an adequate bibliographical review. The experimental part is clearly presented and explained. The figures and their legends make it easy to analyze the data.

This interesting paper is based on both animal and cell models, and the methods used are suitable for such a study. Although this work clearly demonstrates that an excess of ATRA causes a decrease in the expression and cellular level of both RARa and Pitx2 proteins, it suffers a conceptual limitation which means that some of the conclusions and interpretations formulated by the authors are a little too assertive.

Following are a few points that need to be clarified or improved to make the article more suitable for publication.

Major points:

1/ Figure 2 - for experiments based on L6 cells, the authors use ATRA at 10µM and Am580 at 20nM. Both ligands are high-affinity agonists for RARa, with Am580 having the best Kd (approx. 1 Log difference). Such concentrations pose a problem for the interpretation of results. Given the respective affinities of ATRA and Am580 for RARa, it is likely that when the two molecules are combined for cell treatment, ATRA is bound to RARa and, by competition, prevents binding of Am580. In this case, the observed effect of Am580 could not be attributed to RARa activation. However, despite the competition between ATRA and Am580 to bind RARa, if a fraction of RARa is bound to and activated by Am580, why would ATRA used alone not cause the same effects as Am580, since ATRA is a RARa agonist like Am580?

Taken together, these considerations make the mechanistic interpretation of the results presented in figure 2 difficult. Without calling the results into question, the mechanisms proposed by the authors should be reassessed.

Minor points:

2/ Did the authors test ATRA at concentrations below 10µM in the assays presented? Defining whether a lower concentration of ATRA is capable of mediating comparable effects could be informative about ATRA's mechanism of action. Also it should be noted that a concentration of 10µM, ATRA can bind to other cellular targets (PPARs, RORs, others) and regulate pathways controlled by these proteins.

3/ A contradiction is observed between Fig2c and Fig2e. The western blot shows an increase in Pitx2 levels (Fig2c) for ATRA-Am580 co-treatment compared with untreated, whereas quantification of this western blot indicates a decrease.

4/ Why is Am580 not used on its own (Fig. 2)? This experimental condition would constitute a relevant control.

5/ Another point is that in vitro a fraction of ATRA can be converted to 9cRA, an agonist for RARs but also for RXR, the heterodimerization partner of RARs. Simultaneous activation of both RAR and RXR partners by an agonist results in transcriptional over-activation, which can bias assays. On the other hand, as RXR is also the heterodimerization partner of other nuclear receptors, its activation by an agonist like 9CRA is likely to affect these other signaling pathways. It would be advisable to check whether 10µM of the ATRA solution used for these studies does not exhibit RXR agonist activity.

6/ Am580 is considered a selective agonist for RARa over RARb and RARg. However, to respect this specificity, it is necessary to be cautious about the amount of ligand used. It is necessary not to exceed a concentration of 10nM of Am580. 20nM is a little too strong. For experiments using cells in culture, 5nM Am580 is sufficient to fully activate RARa.

7/ In the discussion section, authors mention the response elements (RARE) that enable RARs to associate with DNA in regions regulating target gene expression. They write “RarE sequences”. This term and abbreviation should be defined for readers unfamiliar with this field.

8/ Authors write “chemically synthesized shRNA”. What does the term “chemically” refer to in this context?

9/ Authors use trivial name for RARalpha as well for other nuclear receptors. A logical numbering system and receptor code, supporting the trivial names, was made by the International Committee of Pharmacology Committee on Receptor Nomenclature and Classification (NC-IUPHAR). In each manuscript dealing with nuclear receptors, it is recommended that the receptors be identified by the official names at least once in the summary and the introduction. Once the name has been established, authors may use the trivial name for the remainder of the manuscript. For instance, the trivial name and the formal nomenclature for RARalpha is NR1B1.

Reviewer #2: The authors examine the role of Rara and pitx2 in muscle development, focusing on anorectal malformation caused by faulty development of pelvic floor muscles. Sprague-Dawley rats were given all-trans retinoic acid (ATRA) as a model for anorectal malformation. PFMs of the ARM model rats were poorly developed, and displayed lower expression of myod, mymk, (consistent with faulty development), pitx2, and rara, while pax7 was elevated. Additional experiments in ATRA-treated L6 myoblasts allowed more mechanistic examination. ATRA-treated L6s recapitulated the results seen in PRMs, with decreased levels of pitx2, rara, myod, and mymk. Transfection with Ad-Pitx2 recovered much of these decreases, with the exception of Mymk. These results together indicate that ATRA inhibits L6 differentiation through decreasing Rara and pitx2. Further, siRNA of Rara decreased the expression of pitx2, myod, and mymk. Taken together, these results suggest a potent role for Pitx2 and Rara in muscle differentiation, with relevance to ARM.

The experiments are well performed and presented, with the appropriate statistical analyses. The use of ATRA as a model for ARM and studying myoblast differentiation provides a useful approach that is complemented well by the use of Rara siRNA. These findings suggest a strong role for the Pitx2/Rara axis.

The most exciting experiment remains to be done. Having shown the strong effects of siRara on MyoD levels, and the ability of Pitx2 to restore MyoD and Mymk, analyzing the differentiation of L6 myoblasts in these contexts (as in Fig. 3b and c) would provide strong support for the authors' conclusions and strengthen the manuscript.

As a standard procedure for all manuscripts, preliminary checks indicate that AI was not used in generating the manuscript. The reviewer thanks the authors for their time and integrity.

.

Reviewer #1: No

Reviewer #2: No

While revising your submission, please upload your figure files to the Preflight Analysis and Conversion Engine (PACE) digital diagnostic tool, https://pacev2.apexcovantage.com/. PACE helps ensure that figures meet PLOS requirements. To use PACE, you must first register as a user. Registration is free. Then, login and navigate to the UPLOAD tab, where you will find detailed instructions on how to use the tool. If you encounter any issues or have any questions when using PACE, please email PLOS at . PACE helps ensure that figures meet PLOS requirements. To use PACE, you must first register as a user. Registration is free. Then, login and navigate to the UPLOAD tab, where you will find detailed instructions on how to use the tool. If you encounter any issues or have any questions when using PACE, please email PLOS at . PACE helps ensure that figures meet PLOS requirements. To use PACE, you must first register as a user. Registration is free. Then, login and navigate to the UPLOAD tab, where you will find detailed instructions on how to use the tool. If you encounter any issues or have any questions when using PACE, please email PLOS at . PACE helps ensure that figures meet PLOS requirements. To use PACE, you must first register as a user. Registration is free. Then, login and navigate to the UPLOAD tab, where you will find detailed instructions on how to use the tool. If you encounter any issues or have any questions when using PACE, please email PLOS at figures@plos.org. Please note that Supporting Information files do not need this step.. Please note that Supporting Information files do not need this step.

---

## [Author Response · Author response to Decision Letter 1]

31 Dec 2025

Dear Editor of PLOS one:

We sincerely thank the editor and all reviewers for their valuable feedback that we have used to improve the quality of our manuscript. The reviewer comments are laid out below in italicized font and specific concerns have been numbered. Our response is given in normal font and changes/additions to the manuscript are given in the red text.

Reviewer #1:

Major points:

1/ Figure 2 - for experiments based on L6 cells, the authors use ATRA at 10µM and Am580 at 20nM. Both ligands are high-affinity agonists for RARa, with Am580 having the best Kd (approx. 1 Log difference). Such concentrations pose a problem for the interpretation of results. Given the respective affinities of ATRA and Am580 for RARa, it is likely that when the two molecules are combined for cell treatment, ATRA is bound to RARa and, by competition, prevents binding of Am580. In this case, the observed effect of Am580 could not be attributed to RARa activation. However, despite the competition between ATRA and Am580 to bind RARa, if a fraction of RARa is bound to and activated by Am580, why would ATRA used alone not cause the same effects as Am580, since ATRA is a RARa agonist like Am580?

Taken together, these considerations make the mechanistic interpretation of the results presented in figure 2 difficult. Without calling the results into question, the mechanisms proposed by the authors should be reassessed.

Answer1: In our experiment, we initially used high concentrations of ATRA to stimulate L6 cells, followed by a switch to differentiation medium containing AM580 to further stimulate the cells. Previous studies have shown that the biological effects of ATRA are highly concentration-dependent. Excessive exposure to ATRA can not only reduce RARα expression levels but also promote ubiquitination and degradation of the RARα protein by activating the proteasome pathway, thereby exerting negative feedback inhibition on the signaling pathway. Therefore, after high concentrations of ATRA caused a decrease in RARα levels in L6 cells, subsequent stimulation with AM580 enhanced the effects of ATRA on these cells by acting on the remaining RARα.

Minor points:

2/ Did the authors test ATRA at concentrations below 10µM in the assays presented? Defining whether a lower concentration of ATRA is capable of mediating comparable effects could be informative about ATRA's mechanism of action. Also it should be noted that a concentration of 10µM, ATRA can bind to other cellular targets (PPARs, RORs, others) and regulate pathways controlled by these proteins.

Answer2: We tested lower concentrations of ATRA (<10 nM) and found that it can increase RARα expression. This study was based on human specimens from previous research. Earlier experiments did not detect changes in RARγ and RARβ expression in the terminal rectum of children with ARM. Therefore, this study focused solely on RARα expression. This represents a limitation of the current study, and we plan to conduct more comprehensive experiments in the future.

3/ A contradiction is observed between Fig2c and Fig2e. The western blot shows an increase in Pitx2 levels (Fig2c) for ATRA-Am580 co-treatment compared with untreated, whereas quantification of this western blot indicates a decrease.

Answer3: The result shown in Figure 2e is based on the ratio of the target protein expression level to the GAPDH expression level depicted in Figure 2c. Visually, the concentration of the Pitx2 band in the untreated group in Figure 2c appears low; however, the GAPDH level in the same sample is also low, which explains this outcome.

4/ Why is Am580 not used on its own (Fig. 2)? This experimental condition would constitute a relevant control.

Answer4: As described in Answer 1, AM580 stimulation was applied to L6 cells only after ATRA stimulation; therefore, we did not include results for L6 cells stimulated with AM580 alone.

5/ Another point is that in vitro a fraction of ATRA can be converted to 9cRA, an agonist for RARs but also for RXR, the heterodimerization partner of RARs. Simultaneous activation of both RAR and RXR partners by an agonist results in transcriptional over-activation, which can bias assays. On the other hand, as RXR is also the heterodimerization partner of other nuclear receptors, its activation by an agonist like 9CRA is likely to affect these other signaling pathways. It would be advisable to check whether 10µM of the ATRA solution used for these studies does not exhibit RXR agonist activity.

Answer5: We will continue the experiment to detect the expression level of RXR in the terminal rectum of children with ARM and to investigate the effect of ATRA on RXR expression in L6 cells.

6/ Am580 is considered a selective agonist for RARa over RARb and RARg. However, to respect this specificity, it is necessary to be cautious about the amount of ligand used. It is necessary not to exceed a concentration of 10nM of Am580. 20nM is a little too strong. For experiments using cells in culture, 5nM Am580 is sufficient to fully activate RARa.

Answer6: The concentrations of ATRA and AM580 used in this study were evaluated using the CCK-8 assay to determine their effects on L6 cells, and the IC50 values were calculated. Subsequently, we will test the effects of varying concentrations of AM580 on L6 cells.

7/ In the discussion section, authors mention the response elements (RARE) that enable RARs to associate with DNA in regions regulating target gene expression. They write “RarE sequences”. This term and abbreviation should be defined for readers unfamiliar with this field.

Answer7: Already noted in the article.

8/ Authors write “chemically synthesized shRNA”. What does the term “chemically” refer to in this context?

Answer8: The shRNA was designed and synthesized by a third-party company; therefore, the related description has been removed from the article.

9/ Authors use trivial name for RARalpha as well for other nuclear receptors. A logical numbering system and receptor code, supporting the trivial names, was made by the International Committee of Pharmacology Committee on Receptor Nomenclature and Classification (NC-IUPHAR). In each manuscript dealing with nuclear receptors, it is recommended that the receptors be identified by the official names at least once in the summary and the introduction. Once the name has been established, authors may use the trivial name for the remainder of the manuscript. For instance, the trivial name and the formal nomenclature for RARalpha is NR1B1.

Answer9: The corresponding content has been added to the abstract.

Reviewer #2:

The most exciting experiment remains to be done. Having shown the strong effects of siRara on MyoD levels, and the ability of Pitx2 to restore MyoD and Mymk, analyzing the differentiation of L6 myoblasts in these contexts (as in Fig. 3b and c) would provide strong support for the authors' conclusions and strengthen the manuscript.

Answer1: Because the only non-fluorescent plasmid or adenovirus we customized is Ad-Pitx2, we only performed immunofluorescence experiments on cells treated with ATRA+Ad-Pitx2.

I would like to take this opportunity to once again express my sincere gratitude to the editors and reviewers, and I extend my best wishes for your continued happiness and well-being!

Sincerely Yours,

Dr. Yi Wang

---

## [Decision Letter · Decision Letter 1]

10 Mar 2026

ATRA influences the differentiation and fusion of myoblasts by regulating Rarα/Pitx2, leading to abnormal development of the pelvic floor muscles (PFMs) in fetal rats.

PONE-D-25-14533R1

Dear Dr. Wang,

We’re pleased to inform you that your manuscript has been judged scientifically suitable for publication and will be formally accepted for publication once it meets all outstanding technical requirements.

Kind regards,

Michael Schubert

Academic Editor

PLOS One

Reviewers' comments:

Reviewer's Responses to Questions

**Comments to the Author**

Reviewer #1: All comments have been addressed

Reviewer #2: (No Response)

2. Is the manuscript technically sound, and do the data support the conclusions?

Reviewer #1: Yes

Reviewer #2: (No Response)

3. Has the statistical analysis been performed appropriately and rigorously?

Reviewer #1: Yes

Reviewer #2: (No Response)

4. Have the authors made all data underlying the findings in their manuscript fully available?

Reviewer #1: Yes

Reviewer #2: (No Response)

5. Is the manuscript presented in an intelligible fashion and written in standard English?

Reviewer #1: Yes

Reviewer #2: (No Response)

Reviewer #1: I would like to thank the authors for their responses and arguments, which have helped to clarify several points in this article. This work now deserves to be continued in order to better define the mechanisms of action involved.

Reviewer #2: (No Response)

.

Reviewer #1: No

Reviewer #2: No

---

## [Editor Report · Acceptance letter]

PONE-D-25-14533R1

PLOS One

Dear Dr. Wang,

I'm pleased to inform you that your manuscript has been deemed suitable for publication in PLOS One. Congratulations! Your manuscript is now being handed over to our production team.

Kind regards,

on behalf of

Dr. Michael Schubert

Academic Editor

PLOS One